# Psychiatric Morbidity, Utilization and Quality of Mental Health Care in Long-Term Unemployed People

**DOI:** 10.3390/ijerph20065066

**Published:** 2023-03-13

**Authors:** Birgit Meiler, Andreas G. Franke, Norbert Scherbaum, Josef Rabl

**Affiliations:** 1Department of Psychiatry and Psychotherapy, LVR-University Hospital Essen, University of Duisburg-Essen, Virchowstraße 174, 45147 Essen, Germany; 2University of Applied Labour Studies, Seckenheimer Landstraße 16, 68163 Mannheim, Germany; 3Johannesbad Kliniken Fredeburg GmbH, Zu den drei Buchen 1, 57392 Schmallenberg, Germany

**Keywords:** unemployment, psychiatric morbidity, utilization, mental health care

## Abstract

Research has shown complex interactions between unemployment and mental health. However, the prevalence of specific mental disorders, utilization of mental health care services and influences on help-seeking behavior have been investigated surprisingly little in the past. In this study, we investigated a sample of long-term unemployed people in a cooperation program of the local unemployment agency and a psychiatric university hospital in a larger city in Germany. Mental disorders, treatment history, accordance of treatment to national treatment guidelines and factors influencing previous treatment were assessed. Participants (*n* = 879; male 56%, female 44%, mean age 43.9 years) showed a high psychiatric morbidity, mostly with diagnoses from the ICD-10 categories F1 (22%), F3 (61%) and F4 (68%). Currently, 18% were in psychiatric treatment, 6% were in psychotherapeutic treatment, and 28% received psychopharmacological treatment. Mostly young men underutilized the psychiatric–psychotherapeutic system, with middle-aged men and women being most frequently in psychopharmacological treatment. Of those treated, only about 10% of the subjects currently received a treatment according to national guidelines. The utilization of psychotherapeutic treatment was strikingly poor. This study identified high psychiatric morbidity and severe treatment gaps in unemployed people. These results can help to target subjects with specific needs for interventions and to modify counseling programs.

## 1. Introduction

The complex relationship between unemployment and health impairments has been widely studied, specifically regarding unemployment and mental illness. Research has shown bidirectional interactions, wherein the causation hypothesis assumes that unemployment leads to mental illness through psychosocial stressors and an increase in risk behaviors, and the selection hypothesis postulates that mental illness increases the risk of losing a job and remaining unemployed [1,2,3,4,5]. It has been shown that the unemployed have up to two times an elevated risk of suffering from mental illness compared to those employed [6,7]. The causation hypothesis assumes that stressors associated with unemployment, such as a loss of daily structure, social participation and self-efficacy, generate psychosocial stress that can promote health risk behaviors (e.g., substance use) and the occurrence of mental illness [7]. Meta-analyses of longitudinal studies have found deteriorations in mental health with small to medium effect sizes following job loss and an improvement in mental health with medium effect sizes following reentry to the job market [8]. The selection hypothesis postulates that people with mental illness are more likely to lose their jobs due to absenteeism, less flexibility and reduced performance. Due to the psychological restrictions, such as lack of drive or reduced self-esteem, as well as self- and external stigmatization processes, reintegration into the labor market is also more difficult, which impairs those affected, for example, in job application requirements [9]. The study of causal and selective processes in the context of unemployment and mental illness is important, as it could provide information for the development of prevention measures, low-threshold integrated care services and targeted health promotion programs. To date, surprisingly few studies have explored the prevalence of mental disorders and factors influencing mental health care utilization in unemployed people in detail. However, this would be important not only to ascertain adequate treatment but also regarding targeted job-related support. This is specifically true, as the return to the labor market is more difficult for people affected by mental illness [10,11,12]. The degree of utilization and quality of disorder-specific treatment is an important factor influencing the course of the disease and thus the likelihood of returning to the labor market. Studies on the utilization behavior of people suffering from mental disorders in the general German population showed an overall low access to mental health care [13]. Several studies stated utilization of the health care system in general and mental health care in particular to be higher among unemployed persons and those with psychiatric multimorbidity [13,14]. Kraut et al. found that any history of mental illness in the past led to persistently higher use of the health care system, irrespective of employment status [15].

Considering these findings, programs to determine and promote mental health among the long-term unemployed have been developed in recent years in a cooperative effort between the unemployment agency of the city of Essen, Germany (Job Center Essen) and the Department of Psychiatry and Psychotherapy of the LVR-University Hospital Essen [16,17]. In this study, we aimed not only to assess current mental disorders in the long-term unemployed, but also the treatment history regarding especially the utilization of mental health services and the quality of treatment evaluated according to the respective treatment guidelines. In addition, specific risk groups and factors influencing utilization behavior were investigated. We hypothesize a treatment gap concerning guideline-based mental health care and assume that specific subgroups in the population of the long-term unemployed differ in utilization, which is why we subdivided the sample according to the specific variables of sex, age, social relations and education level. This specification is important, as it can determine specific risk groups that should be particularly addressed regarding mental healthcare and specific job-related interventions.

## 2. Materials and Methods

The data collected for this study were obtained as part of a collaboration between the unemployment agency of the city of Essen (Job Center Essen) and the LVR-University Hospital, Department of Psychiatry and Psychotherapy at the University of Duisburg-Essen. With almost 600,000 inhabitants, Essen is currently the tenth largest city in Germany, with about 30,000 unemployed people in 2022. We included long-term (at least one year) unemployed people aged 20–66 years who reported mental health issues at the unemployment agency or were perceived by their case managers to be experiencing psychological distress. The case managers referred these clients to an outpatient service at the LVR-University Hospital to participate in a program called “Fit for Work”, which the Department for Psychiatry and Psychotherapy conducts in cooperation with a local provider of vocational training and integration.

Diagnostic procedures included the completion of self-report questionnaires, such as the Beck Depression Inventory (BDI-II) [18] and the patient questionnaire of the Structured Clinical Interview for Personality Disorders (SCID-5 PD) [19]. The Diagnostic Interview for the Diagnosis of Psychiatric Disorders (Mini-DIPS) [20,21] was carried out with all subjects. If the patient questionnaire of the SCID-5-PD revealed evidence of the presence of a personality disorder, the clinical interview for the diagnosis of personality disorders (SCID-5-PD) was additionally administered. Diagnostic testing was performed during 4–6 appointments with clinical psychologists. Socio-demographic data and the medical history were collected with a semi-structured self-questionnaire. Concerning treatment history, current and previous utilization of psychiatric, psychopharmacological and psychotherapeutic treatments were recorded. The assessment of whether participants reported treatment was guideline-compliant and classified based on subjects’ self-reported information about medication (concerning substance and dose), as well as psychiatric or psychotherapeutic treatment compared to national treatment guidelines of the respective disorders [22,23,24,25,26]. The distinction between psychiatric and psychopharmacological treatment was made to differentiate the rate of psychiatrically prescribed medication from that of primary care prescriptions.

Statistical analysis was conducted using Rstudio, Posit Software, Boston, USA version 2022.12.0 and SPSS Statistics for Windows, IBM, New York, USA, Version 28.0. We calculated the odds ratios with 95% confidence intervals (CI) and tested group differences for significance using Chi-square tests, supplemented by Fisher’s exact test in the case of small samples. Pearson’s product moment correlations were calculated accordingly, with 95% confidence intervals. We also calculated descriptive measures such as means and standard deviations. To identify specific risk factors for low utilization of health services, independent variables such as gender, age (subdivided into three age groups: 20–35, 36–50 and 51+ years), marital status, level of education, employment history, migration background and financial burdens were each examined by means of odds ratios in comparison to the total sample. For each mental health care category (psychiatric treatment, psychopharmacological treatment and psychotherapy), two dichotomous outcome variables (use versus no use) were defined.

## 3. Results

### 3.1. Demographic Data

We included 879 subjects aged 22 to 66 years in the study. The mean age was 43.9 years (standard deviation (sd) = 10.2. years). Of the participants, 44% were female, and 56% were male. Eighty-five percent of the subjects (*n* = 753) had German citizenship; the other participants had a history of migration but sufficient understanding and command of the German language to participate in the diagnostic process. From 846 subjects who provided information on their marital status, 63% reported living alone, corresponding with a high rate of participants who were not living in a relationship (53.3%). Only 27% of the subjects declared being married or in a steady relationship. Concerning educational qualifications, 10% of the participants had not completed school, 37% had a special school or general certificate of education, 30% had an intermediate school-leaving certificate, 22% had a general or subject-related university qualification and 1% completed university. The levels of educational qualification were nearly equal for both sexes regarding school dropouts and lower educational levels. Higher educational levels were more often found in males (24% versus 16% in females). A total of 45% of the participants reported having completed vocational training. Data on the history of unemployment were collected from 675 subjects, from which 530 subjects reported having been employed at least once in their lifetime, and 460 subjects provided information on how long they had been unemployed. The average duration of unemployment was 91 months (sd = 77.4 months), with women being unemployed longer on average. A total of 22% (*n* = 145) of the participants reported that they had never been employed. There was a positive correlation between the duration of unemployment and age of r(458) = 0.34, *p* < 0.001.

### 3.2. Diagnostic Data

We collected diagnostic data from 878 subjects who completed the diagnostic procedure. Ninety-three percent (*n* = 811) of the participants were diagnosed with at least one mental disorder. Two-thirds (*n* = 529) of those examined had at least two diagnoses.

The most frequently diagnosed disorders were from ICD-10 [27] categories F1 (mental and behavior disorders due to psychoactive substance abuse), F3 (affective disorders) and F4 (neurotic, stress-related and somatoform disorders). In the F1 category, alcohol dependence and cannabis dependence were the most frequent diagnoses, with the former being diagnosed most frequently among older men and the latter among younger men (see Table 1). Overall, addictive disorders were far more common in men than in women (28% versus 16%).

Affective disorders were diagnosed at about the same rate in both sexes and affected two-thirds of participants. Here, recurrent episodes of a major depression were diagnosed five times more frequently than single depressive episodes. In comparison, women were more likely to have recurrent depressive disorders than men (38% versus 30%). Diagnoses in the F4 category were 15% more common among female study participants than among males (66% versus 51%). The high frequency of social phobias, particularly in the younger and middle age groups, is striking. Among young women and middle-aged men, approximately 30% were affected. Posttraumatic stress disorders (PTSD) were the second most common diagnosis from the F4 category after social phobia, affecting 20% of all subjects. Here, women were affected almost three times more frequently than men (29% versus 13%). Personality disorders were diagnosed in about 10% of subjects. Here, men were affected slightly more often (12% versus 8.5%). Emotionally unstable personality disorder (2.5%) and anxious avoidant personality disorder (1.8%) were the most common disorders in the F6 category. Among comorbidities, the combination of a moderate episode of major depression and a disorder from the F4 category were most common, with social phobia being by far the most common comorbid disorder (13% of all comorbidities), followed by agoraphobia and PTSD. Comorbid substance-related dependence disorders were also common, accounting for 10% of all comorbidities.

### 3.3. Health Care Utilization

Regarding health care utilization, *n* = 854 subjects provided information on current and past psychiatric, psychopharmacological or psychotherapeutic treatment (see Table 2). At the time of participation, only 18% were in current psychiatric treatment, 28% had psychopharmacological treatment (probably partly prescribed by a general practitioner), and only 5% had psychotherapeutic treatment.

Men used fewer treatment options overall, with the gap in care being particularly pronounced among young men (see Table 3). Across all participants, the middle-aged group used all three treatment options most frequently, except for medication, which was most often used by older women. Women in the middle-aged group (36–50 years) most frequently used psychotherapeutic treatment. There was no significant relationship between the number of diagnoses and the utilization of psychiatric or psychotherapeutic treatment. However, there was a significant correlation between the use of psychopharmacological treatment and psychiatric morbidity, measured by the number of comorbid mental disorders (r(859) = 0.12, *p* < 0.001). Just under half of the subjects had used psychiatric treatment or medication in the past, and only 30% had psychotherapeutic treatment. About half of the participants reported that their last health care utilization was more than 5 years ago.

The most commonly prescribed psychotropic drugs were antidepressants (64%), with selective serotonin reuptake inhibitors (SSRI) and serotonin–noradrenaline reuptake inhibitors (SNRI) being the most common medication. Both SSRIs and SNRIs were prescribed mostly for recurrent depressive disorders, single depressive episodes and social phobia. The second and third most commonly prescribed psychotropic drugs were neuroleptics (17%) and sedatives (13%), with neuroleptics most commonly used for major depression and sedatives for both anxiety and depressive disorders.

Polypharmacy (more than three drugs prescribed) was rarely found in this sample. The frequent prescription of sedating substances was striking, among which tricyclic piperazinyl derivates (opipramol) were most frequently taken (8.5% of all medication).

The comparison of drug treatment with national treatment guidelines showed that on average, only slightly more than 50% of drug therapies were compliant with the guidelines (see Table 4). In both sexes, the proportion with guideline-compliant pharmacotherapy was higher in the younger age group (20–35 years: 59.5%) than in the older age group (>50 years: 46.4%). The self-reported compliance concerning medication was nearly equal, around 80% in both sexes and the three age groups.

The analysis of the probability of using one of the three treatment options showed partly significant differences with regard to sex, certain age groups, as well as the level of education (see Table 5). Males in the younger age group (20–35 years) were significantly less likely to be in psychiatric treatment or taking medication (OR 0.4, *p* < 0.05). Males in the middle-aged group, on the other hand, were more likely to be in psychiatric treatment (OR 1.6, *p* < 0.05), and women in the same age group were significantly more likely to be in psychotherapy (OR 2.4, *p* < 0.05) or taking medication (OR 1.6, *p* < 0.05); hence, being middle-aged is a favoring factor for the utilization of mental health care. In terms of educational attainment, subjects with intermediate levels of education were found to be less likely to receive psychiatric treatment (0.6, *p* < 0.05), and subjects with high levels of education were found to be more likely (1.6, *p* < 0.05). Marital status and migration background had no significant effect on mental health care utilization. In summary, female sex, higher educational attainment and a stable partnership were associated with a higher likelihood of using mental health care.

## 4. Discussion

In the present study, we describe the results of a large cohort of long-term unemployed people investigated within a program to identify and promote mental health in an urban setting in Germany. The study finds high levels of psychiatric morbidity in this sample. The spectrum of diagnoses differed regarding distribution from the prevalence rates in samples from the general population and cohorts from studies of health insurers [13,28]. Diagnoses from categories F0 (organic, including symptomatic mental disorders), F2 (schizophrenia, schizotypal and delusional disorders) and F7 (mental retardation) were rare in the cohort studied, because people affected by mental illnesses in this category presumably often lose access to the labor market entirely or work in sheltered employment. Particularly striking in our study was the accumulation of social phobias (23%), PTSD (20%) and chronic depressive disorders (33%). In the German National Health Survey 2012 (10), simple depressive episodes (F32 10.6%) were more common than recurrent episodes of major depression (F33 8.4%). Among anxiety disorders, agoraphobia was the most common (5.6%), and social phobias and PTSD were equally common at 3.6%. However, unlike our sample, which examined individuals who reported psychological distress, the National Health Survey examines a representative sample of the general population. In addition, we found a high percentage of substance-related dependence, especially in men, as a comorbidity to major depression.

Although the burden of psychiatric morbidity was high in the sample, we found a contradicting low utilization of mental health care. Psychotherapy in particular was dramatically underutilized. This is particularly important in light of the fact that national and international treatment guidelines recommend psychotherapeutic treatment as a first-line therapy for the most frequently diagnosed disorders (social phobia, PTSD and major depression) with high levels of evidence [22,23,26]. According to reporting from a 2014 National Health Interview Survey [29], around 10% of the adult population reported at least one contact with a psychiatrist or psychotherapist in the past 12 months. Depressed individuals had sought psychiatric or psychotherapeutic help at a rate of over 30% in the same survey. In the German general population, the use of psychiatric–psychotherapeutic treatment increases with age, with a maximum percentage around the age of 55 [29]. Former findings that women, those in committed relationships and the formerly employed were more likely to be in treatment were confirmed. Contrary to other studies that showed higher medical care utilization among cohorts with low education, this study showed the opposite picture. Taking into account the high burden of mental illness in the sample studied, our study shows a clear underuse, especially with regard to psychotherapeutic treatment. The costs of outpatient psychotherapy, psychiatric treatment and medication are covered by statutory health insurance funds in Germany. Access is therefore basically independent of financial means or employment status. One could hypothesize that the high number of subjects diagnosed with social phobia may contribute to the poor utilization of psychotherapy. In particular, among younger people, social phobia was diagnosed in 30% of the subjects in our sample in this age group. Entering psychotherapeutic treatment presents a social demand that could be particularly challenging for socially phobic individuals. In addition, in the sample studied here, the number of subjects with dependency disorders was high, which is a barrier to admission to outpatient psychotherapy in the German health care system.

At first glance, psychopharmacological treatment seems comparatively good, but only about half of the drug treatments (13%) were compliant with national guidelines. The overall quality of treatment as defined by the respective disorder-specific treatment guidelines was poor in the cohort studied. The treatment guidelines for anxiety disorders primarily recommend psychotherapeutic treatment and only secondarily drug treatment, especially with selective serotonin reuptake inhibitors or serotonin–noradrenaline reuptake inhibitors. Similarly, in the national treatment guidelines for PTSD, there is a first-line recommendation for trauma-focused psychotherapy.

A limitation of the present study is that participants were pre-selected by assignment from the unemployment agency’s case managers. The selection criterion was that the participants themselves reported psychological stress or appeared to the case managers to be psychologically stressed. The number of those who met the selection criteria but declined to participate in the program was not available to us. In addition, it would be useful to examine a sample of unselected long-term unemployed persons. Another limitation of the study is that the case numbers of those who used psychotherapy within the total cohort were so small. Hence, it was not possible to identify positive factors influencing the use of guideline-based psychotherapy with sufficient statistical power. In addition, a limitation could arise from the fact that the collection of treatment data was based on self-report by the subjects.

## 5. Conclusions

In summary, this study highlights the urgent need for targeted, guideline-based treatment among long-term unemployed individuals with mental illness. The risk factors described here, such as social isolation, frequent use of addictive substances and the significantly increased prevalence of social anxiety disorders as well as PTSD compared to the general population, understandably inhibit access to treatment services. Avoidance behaviors, fear of social judgement and re-actualization of stressful life events are significant barriers to mental health care in the aforementioned disorders. Self-stigmatization and stigmatization by others also play a role here, both because of the mental illness and because of unemployment [30]. Especially in the younger age group, it can be assumed that the transition from the school system to the labor market is interrupted by mental disorders. Particularly for this age group, low-threshold access to psychiatric and psychotherapeutic care in integrated care concepts would be urgently needed both to prevent the chronification of mental disorders and to maintain employment perspectives. Our findings underscore the need for programs that accompany both general social activation and low-threshold transition into guideline-based treatment services. Cooperation programs between unemployment agencies and mental health care institutions that ensure guideline-based treatment could greatly facilitate access to psychiatric–psychotherapeutic care for the long-term unemployed and improve its quality. A large proportion of the mental disorders diagnosed in this study have a favorable course under psychotherapeutic treatment and adequate medication. The societal costs of a treatment gap in the cohort studied are high, which is why there is an urgent need for action.

## Figures and Tables

**Table 1 ijerph-20-05066-t001:** Sociodemographic data.

		All (*n* = 879)	Female (*n* = 388)	Male (*n* = 490)	X^2^	*p*
Age	Mean, (SD)	43.91, (10.14)	44.14, (10.14)	43.78, (10.16)		
Citizenship	German	753	323	430	4.8046	0.028
	Other	108	59	49		
History of Migration	Yes	133	67	66	2.1352	0.144
	No	714	308	406		
Marital Status	Married	67	30	37	0.000	0.989
	Single	452	149	302	52.856	0.000
	Cohabitant	159	91	68	11.832	0.000
	Living separated	31	16	15	0.36656	0.544
	Divorced	125	82	43	24.942	0.000
	Widowed	13	10	3	4.3029	0.038
Level of Education	Not finished school	84	35	49	0.11092	0.739
	Special school	19	12	7	2.1639	0.141
	General certificate of education	288	141	147	4.1729	0.041
	Intermediate school-leaving certificate	243	107	135	0.000	0.989
	Subject-related university qualification	99	32	67	5.6732	0.017
	General university qualification	79	29	50	1.5591	0.211
	University degree	8	4	4	0.000	0.989
Ever employed	Yes	530	241	289	2.5265	0.282
	No	145	72	73		
Time of unemployment	Mean, (SD)	90.89, (77.39)	98.28, (82.65)	85.22, (72.97)		

Note: sociodemographic data of the study sample, with chi-square tests on significant differences in gender.

**Table 2 ijerph-20-05066-t002:** Frequencies (%) of psychiatric diagnoses in sexes and age groups (in years) subdivided in ICD-categories and most frequently diagnosed disorders of the respective category.

	All	Female				Male					
	(*n* = 878)	(*n* = 388)				(*n* = 490)					
	All	All	20–35 y *n* = 113	36–50 y *n* = 152	51+ y *n* = 123	All	20–35 y *n* = 127	36–50 y *n* = 215	51+ y *n* = 148	X^2^	*p*
**F1 (all)**	**21.5**	**15.5**	**18.6**	**15.1**	**13.0**	**28.4**	**28.3**	**30.2**	**25.7**	**20.569**	**0.000**
F10.1	3.0	2.1	0.9	2.6	2.4	3.7	3.1	4.2	3.4	1.957	0.162
F10.2	7.4	3.6	2.7	2.6	5.7	10.4	4.7	11.2	14.2	14.606	0.000
F12.1	2.1	0.5	0.9	0.7	0	3.3	3.9	4.2	1.4	8.154	0.004
F12.2	7.4	5.4	8.8	5.3	2.4	9.0	12.6	10.2	4.1	4.020	0.045
**F2 (all)**	**2.1**	**1.5**	**2.7**	**0.7**	**1.6**	**2.4**	**0.8**	**4.2**	**1.4**	**0.878**	**0.349**
**F3 (all)**	**60.5**	**62.6**	**65.5**	**63.8**	**58.5**	**58.8**	**63.8**	**59.5**	**53.4**	**1.345**	**0.246**
F32.0	1.8	2.1	1.8	2.6	1.6	1.6	3.9	0.5	1.4	0.223	0.637
F32.1	5.1	4.1	5.3	3.9	3.3	5.9	8.7	5.1	4.7	1.434	0.231
F32.2	2.2	1.8	2.7	2.0	0.8	2.4	3.1	0.9	1.2	0.425	0.514
F33.0	5.5	6.2	4.4	7.2	2.1	4.9	7.1	4.2	4.1	0.695	0.405
F33.1	17.8	21.9	20.4	21.1	24.4	14.5	13.4	16.7	12.2	8.154	0.004
F33.2	10.0	9.8	5.3	11.8	11.4	10.2	9.4	11.6	8.8	0.040	0.841
F34	10.3	8.0	12.4	5.9	6.5	12.0	14.2	12.6	9.5	3.863	0.049
**F4 (all)**	**58.1**	**65.7**	**68.1**	**69.1**	**59.3**	**51.2**	**58.3**	**50.2**	**46.6**	**18.638**	**0.000**
F40.0	12.8	16.5	15.9	16.4	17.1	9.8	8.7	9.3	11.5	8.731	0.003
F40.1	22.8	21.6	30.1	26.3	8.1	23.7	33.9	24.2	14.2	0.504	0.478
F41	8.8	11.9	12.4	13.2	9.8	6.3	7.1	5.1	7.4	8.274	0.004
F43	20.2	29.1	31.9	27.0	29.3	13.1	10.2	11.6	17.6	34.710	0.000
**F5 (all)**	**5.6**	**8.0**	**7.1**	**9.2**	**7.3**	**3.7**	**3.9**	**2.8**	**4.7**	**7.656**	**0.006**
**F6 (all)**	**10.6**	**8.5**	**11.5**	**9.9**	**4.1**	**12.0**	**11.0**	**13.5**	**10.8**	**2.886**	**0.089**
F60.3	2.5	3.9	7.1	4.6	0	1.4	3.1	1.4	0	5.266	0.022
F60.6	1.8	1.5	0.9	2.0	1.6	2.0	0.8	3.3	1.4	0.296	0.586
**F7 (all)**	**0.5**	**0.3**	**0**	**0.7**	**0**	**0.6**	**1.6**	**0.5**	**0**	**0.600**	**0.439**
**F8 (all)**	**0.9**	**0.3**	**1.0**	**0**	**0**	**0.2**	**0**	**1.0**	**0**	**0.027**	**0.868**
F9 (all)	2.5	0.8	1.8	0.7	0.0	3.3	5.5	3.3	1.4	6.352	0.012

Note. Shown in this table are the ICD-10 Categories of Chapter V (mental and behavioral disorders): F1: mental and behavioral disorders due to psychoactive substance use, F2: schizophrenia, schizotypal and delusional disorders, F3: mood (affective) disorders, F4: neurotic, stress-related and somatoform disorders, F5: behavioral syndromes associated with physiological disturbances and physical factors, F6: disorders of adult personality and behavior, F7: mental retardation, F8: disorders of psychological development, F9: Behavioral and emotional disorders with onset usually occurring in childhood and adolescence. Specific diagnoses are: F10.1: alcohol use disorder (harmful use), F10.2: alcohol dependency, F12.1: cannabis use disorder (harmful use), F12.2: cannabis dependency, F32.0: mild depressive episode, F32.1: moderate depressive episode, F32.2: severe depressive episode without psychotic symptoms, F33.0: recurrent depressive disorder, episode mild, F33.1: recurrent depressive disorder, episode moderate, F33.2: recurrent depressive disorder, episode severe without psychotic symptoms, F34: persistent affective disorders, F40.0: agoraphobia, F40.1: social phobias, F41: other anxiety disorders, F43: reaction to severe stress and adjustment disorders, F60.3: emotionally unstable personality disorder, F60.6: anxious avoidant personality disorder.

**Table 3 ijerph-20-05066-t003:** Mental health care utilization in sexes and age groups in percent.

	All	Female				Male				X^2^	*p*
	%	All	20–35 y	36–50 y	51+ y	All	20–35 y	36–50 y	51+ y		
**Current**											
Psychiatric treatment	18.1	19	17.9	19.3	19.7	17.5	8.2	23.6	16.7	0.234	0.627
Medication	28.3	33.2	24.3	36.5	37.2	24.3	16.3	30	22.6	7.563	0.0059
Psychotherapy	5.5	7.4	4.9	10.3	6.6	4.1	4.1	4.4	3.5	3.911	0.047
**Past**											
Psychiatric treatment	46.8	53.6	55.8	52.9	48	47.4	43.3	47.6	44.4	3.089	0.078
Medication	45.5	56.5	46.9	56.1	48.8	45.2	41	45.3	48.1	4.027	0.044
Psychotherapy	29.9	37.1	35.4	40.3	34.1	26.4	28.3	25.8	21.3	9.551	0.0019

Note: this table shows the current and past mental health care utilization by gender, with chi-square tests on significant differences in gender.

**Table 4 ijerph-20-05066-t004:** Frequency of guideline-compliant drug treatment.

%	All	Female	Male	X^2^	*p*	%	All	Female	Male	X^2^	*p*
	*n* = 176	*n* = 90	*n* = 23	*n* = 37	*n* = 30		*n* = 176	*n* = 90	*n* = 23	*n* = 37	*n* = 30
	**Medication according to treatment guidelines?**										
	All	All	20–35 y	36–50 y	51+ y	All	20–35 y	36–50 y	51+ y		
Yes	53.1	52.2	56.5	54.1	46.7	53.5	64.3	55.6	46.2	1.2154	0.544
No	46.9	47.8	43.5	45.9	53.3	46.5	35.7	44.4	53.8		
	**Medication taken regularly and according to prescription**?										
	All	All	20–35 y	36–50 y	51+ y	All	20–35 y	36–50 y	51+ y		
Yes	83.6	90.8	82.6	82.9	78.8	85.6	73.7	89.8	82.8	0.37429	0.540
No	16.4	9.2	17.4	17.1	21.2	14.4	26.3	10.2	17.2		

Note: the decision as to whether a medication complied with the treatment guidelines was made by comparing the medication reported by participants with the pharmacotherapy (class of substances and combination of psychotropic drugs) and dosage recommended in the respective guideline.

**Table 5 ijerph-20-05066-t005:** Probability of current mental health care utilization.

	Psychiatric Treatment	Psychotherapy	Medication
	OR (95% CI)	OR (95% CI)	OR (95% CI)
**Age (years**)			
**22–35**	**0.58 ***	**0.7**	**0.6 ***
Female	1.0	1.0	0.7
Male	0.4 *	1.0	0.4 *
**36–50**	**1.5 ***	**1.5**	**1.4 ***
Female	1.1	2.4 *	1.6 *
Male	1..6 *	0.7	1.1
**51+**	**1.0**	**0.8**	**1.1**
Femal	1.1	1.2	1.6
Male	0.8	0.6	0.7
**Marital status**			
Married/cohabitant	1.0	0.9	1.7 *
Single	0.8	1.0	0.8
**Migration background**	0.7	0.9	1.2
**Educational qualification**			
No	0.9	0.7	0.7
Low	1.0	1.0	1.1
Middle	0.6 *	0.8	0.7
High	1.6 *	1.7	1.3
No history of employment	0.8	1.2	1.4
Financial debts	1.0	1.1	1.0

OR, odds ratio; CI, confidence interval. * significance level < 0.05 (Chi-square test, Fisher’s exact test). Educational levels: low: special school and general certificate of education; middle: intermediate school-leaving certificate; high: general or subject-related university qualification.

## Data Availability

Complete raw data are not publicly available, but de-identified data could be made available upon reasonable request from the corresponding author.

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
