# Peer review of "Psychiatric Morbidity, Utilization and Quality of Mental Health Care in Long-Term Unemployed People"

_ijerph, 2023, doi:10.3390/ijerph20065066_

Round 1

Reviewer 1 Report

The article "Psychiatric morbidity, utilization and quality of mental health care in long-term unemployed people" aims to assess mental disorders in a German sample of long-term unemployed, their treatment history and quality, and the use they do of mental health service

The study is interesting and the manuscript was pleasant to read. However, I have some comments which are listed below:

Material and methods

- Lines 76-82. The order of administration of the questionnaires is confusing. Was the MINI DIPS administered first and then, if psychopathology was suspected, the SCID-5-PD or from the beginning did participants complete the SCID-5-PD?

- Lines 77-81. Are the questionnaires you used validated in German? If so, please specify in the manuscript.

- Lines 84-85. Could you specify the difference between psychiatric and psychopharmacological treatment? Do you refer to who instituted the treatment (psychiatrist vs. general practitioner)?

- Line 95. Why did you choose this age range? Is it based on previous literature?

- How did you calculate the necessary sample size?

Results

- Standardize in this section the use of numbers written with letters or digits (e.g. line 103, 115...).

- Section 3.1. A table with the sociodemographic variables would be advisable, also comparing if there are significant differences according to gender.

- Line 117. What is the reason for the large dropout in the unemployment response?

- Section 3.2. When writing ICD-10 codes, explain to which categories corresponds (e.g. F1 = Mental and behavioural disorders due to use of psychoactive substances).

- Table 1. Add a footnote with the meaning of each code. To make a easier understand, consider removing columns 2, 3 and 4 from the "All" section, as this values can be inferred from values from gender sections. Also, add the p-value and corrected typified residuals to see in which categories there are significant differences based on gender.

- Table 2. Idem.

- Table 3. Idem.

- Lines 200-202. Add being middle aged as a favouring factor of mental health care utilization. In addition, you state that having had a previous job is a favouring factor, but you do not give this data in Table 4.

Discussion

- When writing ICD-10 codes, explain to which categories corresponds.

-I would like a more elaborate discussion, not only comparing differences with previous studies, but also hypothesizing about the reasons for the results you obtained such as why there is an underuse of psychotherapy in this population (lines 225-226), why people with low education in your sample do not have a higher medical care utilization like in other studies (234-236), or why a large number of prescribed treatments do not conform to guidelines (238-240). In addition, it would be interesting to comment on what the factors for higher likelihood of using mental health care are due to. In addition, it would be interesting to comment on the reasons why the factors of higher likelihood of using mental health care are so.

Reviewer 2 Report

Please see attached

Reviewer 3 Report

This is an interesting paper on the frequency and utilization of mental health services by long term unemployed persons in Germany.

In the introduction the adherence to the national guidelines on mental health care. It would be beneficial for the readers to provide a short description of what these guidelines entail and how the practice for the mental care of the long term unemployed differs.

It would be also interesting to provide some background data on the rate of job finding in long term unemployed people in Essen.

Another point that needs to be considered is whether in the group examined in the study the effect of multiple morbidities was considered with relation to medication compliance.

Finally, how is the cost of mental treatment covered for these people and how easy it is to get access to the mental health services in Essen in terms of waiting times and follow-up visits?

Round 2

Reviewer 1 Report

The authors answered all my questions. Thank you.